# Thixoforming and Rheo-Die-Casting of A356/SiC Composite

**Sukangkana Talangkun \*** , **Charinrat Potisawang and Phuriphut Saenpong**

Faculty of Engineering, Khon Kaen University, Khon Kaen 40002, Thailand; charinrut.ph.56@ubu.ac.th (C.P.); my.emotion2525@gmail.com (P.S.)
**\*** Correspondence: sukangkana@kku.ac.th; Tel.: +66-43362145-6

**Abstract:** This research investigated the rheo-die-casting process and the production of an A356/silicon carbide (SiC) composite by a thixoforming. The composite contained 15 percent by weight SiC particles of around 15–37 μm in size as the reinforcing phase. The composite feedstock was produced by semi-solid stir-casting, where pretreated SiC powder was gradually added into the A356 alloy melt to form a continuously stirred slurry composite melt, which was then cast in a steel mold. For thixoforming, the feedstock was reheated to 583 °C (approximately 0.4 fraction liquid), and its viscosity was reduced with shear rate, implying that A356/SiC exhibits shear thinning or non-Newtonian behavior. This is caused by the characteristic billet structure obtained having relatively globular grains that accommodate the flow of the semisolid composite. In the rheo-die-casting process, the A356/SiC feedstock was re-melted at 610–615 °C (approximately 0.8–0.9 fraction liquid) prior to die-casting and the resulting slurry was injected into a die with injection speeds of 3 and 4 m/s and pressures of 11 and 12 MPa, respectively. Two work-pieces of $16 \times 15.6 \times 205$ mm$^3$ were produced in one shot, and the resulting samples were subjected to T6 solution treatment at 540 °C for 1 h, quenched, and aged at 135 °C for 12 h. The results show that both low speed and low pressure rheo-die-cast samples exhibit uneven filling at the end of the part, whilst both high pressure and high speed promote more uniform distribution of SiC particles throughout the part length. In the as-rheo-die-cast condition, the most uniform of microstructures and hardness obtained from a sample fabricated at 4 m/s speed and 12 MPa pressure.

**Keywords:** A356/SiC composite; rheo-die-casting; thixoforming

## 1. Introduction

Aluminum Matrix Composites (AMCs) have been widely used to date in aerospace and automotive applications. A356 reinforced with ceramic particles, such as silicon carbide (SiC), is one of the major AMCs accepted as an alternative material in automotive parts, especially in brake rotor or disc brakes, as A356/SiC has superior mechanical properties, together with weight reduction of approximately 45 to 60 percent, than conventional cast iron parts [1,2]. Increase in the amount of silicon carbide particles and decrease in the size of silicon carbide particles will improve hardness and consequently improve the compressive strength of Al–SiC composites [3,4]. However, it is crucial that in order to enhance wear resistance, good uniformity of reinforcing particles in the metal matrix and good bonding at the interface, is necessary. Conventional casting, powder metallurgy and semi-solid processing all can produce particulate composites but conventional liquid casting has the disadvantages of segregation and agglomeration of the reinforcing phase and solidification shrinkage, whilst powder metallurgy is limited to small parts. Mechanical stirring, one of the semi-solid processes used, has been shown to improve or avoiding such problems [5]. This simple and effective process can change a dendritic structure to a near-spheroidal grain structure thus promoting a uniform distribution of the reinforcing

particles in the aluminum alloy matrix [6]. Techniques in shaping composites in the semi-solid state include thixoforming and rheocasting. Thixoforming is a semi-solid process that employs slurries at 0.3–0.5 fraction solid and when applied to composites it can produce parts with low porosity and less gas entrapment [7].

Recently, in order to overcome some of the difficulties, such as high porosity, gas entrapment, and flow-ability of the slurry in the die cavity, a Rheo-die-casting technique (RDC), a combination of rheocasting and traditional High Pressure Die Casting (HPDC), has been developed for particle reinforced composite casting in the semi-solid state [8,9]. This technique combines the advantages of a High-pressure die-casting, coupled with semi-solid processing to allow laminar flow, reduced porosity, resulting in uniform and homogenous microstructures. Increasing speed and applied pressure during the solidification stage can also refine intermetallic phases. By comparing A356 formed by thixoforging with rheocasting found that tensile strength increased with decreasing forming temperature, as higher forming temperature increased porosity and distribution of reinforcement phase [10]. Rheocasting combined with HPDC was also successful in forming parts with good distribution of micro to nano size particle phases [11].

The present work focused on a commercially available Al-Si-Mg casting alloy, alloy A356, one of the most widely used heat treatable casting aluminum alloys for parts and components in a variety of industrial applications. The major hardening phase is $Mg_2Si$. After cooling, A356 is usually subject to heat treatment in the T6 condition, a solution treatment used in order to produce a Super Saturated Solid Solution (SSSS), rapid quench in water, and the artificial age hardening to induce small precipitates, scattered throughout the metal matrix. However, when casting semi-solid composites, the semi-solid alloy with embedded ceramic particles cools at greater cooling rates than in conventional casting, implying that the matrix becomes more solution saturated, which in turn requires a different heat treatment regime to conventional liquid and semi-solid cast alloys [10,12]. A long solution time of A356 can be attributed for the decline of Mg solutes used to form $Mg_2Si$ [13].

The present research aimed to assess the thixoformability and rheo-die-casting ability of A356/SiC feedstock produced by a mechanical stir-casting process. In addition, the effects of two major parameters, injection speed and pressure, on the rheo-die-cast microstructure, as well as the distribution of SiC particles in the aluminum A356/SiC composite, are studied. The effects of SiC on aged microstructure and hardness in the T6 condition are also measured.

## 2. Materials and Methods

The matrix material used was commercial A356 aluminum alloy ingots, whilst mechanical stirring produced the A356/SiC composite feedstock. The chemical composition of the A356 alloy, as analyzed by Spectrometry (ARL 3460 OE Spectrometer, Thermo Scientific, Waltham, MA, USA), is shown in Table 1. The silicon carbide powder, with average particle size of 15 and 37 μm, was pre-treated by soaking in 10% HF solution for 24 h, rinsed with water, baked at 100 °C for 24 h, and finally calcined at 900 °C for 4 h. The A356/SiC composites had reinforcement loading of 15 wt%.

**Table 1.** Chemical composition of the A356 specimen used in this study (wt%).

| %Composition | Si | Fe | Cu | Mn | Mg | Zn | Ti | Al |
|---|---|---|---|---|---|---|---|---|
| A356 | 7.16 | 0.11 | 0.01 | 0.00 | 0.36 | 0.01 | 0.13 | Balance |
| ASTM * standard | 6.5–7.5 | ≤0.20 | ≤0.20 | ≤0.10 | 0.25–0.45 | ≤0.10 | ≤0.20 | Balance |

* American Society for Testing and Materials.

The melting range and liquid fraction against temperature were determined by DSC (Differential Scanning Calorimetry) using a HITACHI DSC 7020 (Hitachi High-Tech Science Corporation, Tokyo, Japan). A sample (weighing approximately 6 mg) is heated-up in an alumina pan at 10 °C/min to 700 °C, in nitrogen atmosphere. The curve of fraction liquid against temperature shown in Figure 1

was derived from the heat flow against temperature curve. DSC analysis showed the semi-solid range to be between 530 to 622 °C.

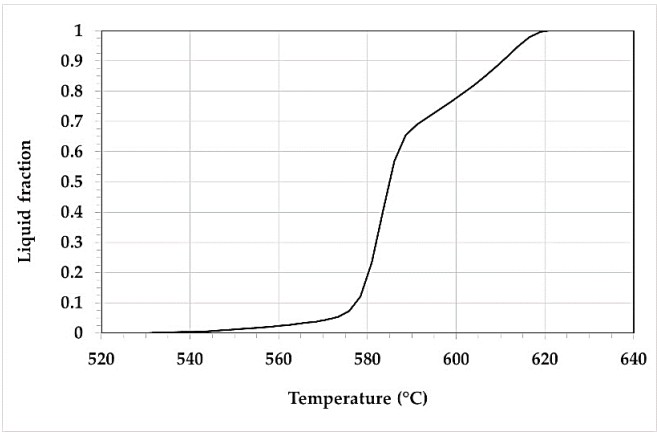

**Figure 1.** Liquid phase versus temperature of A356 obtained by Differential Scanning Calorimetry (DSC).

In order to establish suitable stir casting conditions for A356/SiC, an investigation on the effects of stirring speed and time was initially carried out using only the matrix A356 alloy. The A356 ingot was melted in a graphite crucible by induction, and the near-liquidus slurry (615 °C) of A356 was stirred at 300 and 500 rpm for 5, 10, and 15 min using a stainless double-blade rotor. Once the recommended stirring time was reached, the temperature was rapidly increased again to 680 °C, the impeller removed from the crucible, and the molten composite degassed with argon. Finally, the resulting melt was poured to solidify into a stainless steel mold.

The production of A356/SiC in this experiment, illustrated in Figure 2, has three stages: 1) feedstock production by stir-casting, 2) semi-solid forming by thixoforming, and 3) rheo-die-casting by HPDC.

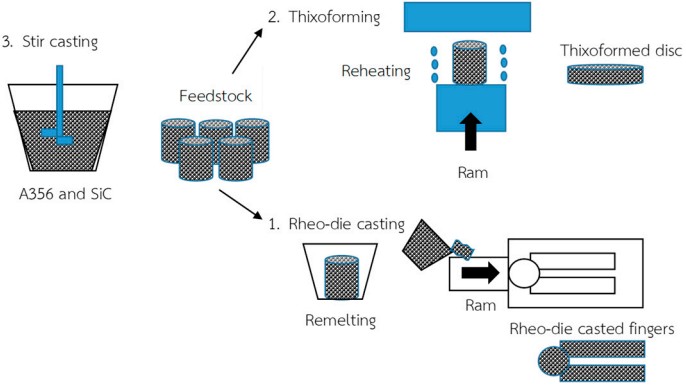

**Figure 2.** Schematic of the experimental set-up. SiC = silicon carbide.

- Stage 1: A356/SiC Feedstock production

Stir casting of A356/SiC follows the preliminary casting of A356 alloy. The stirring time and speed were chosen according to the best microstructure obtained by the preliminary casting of the A356 matrix alloy. For thixoforming, the feedstock is A356/SiC with pretreated silicon carbide size 15 and 37 μm; for the production of the rheo-die-casting feedstock, only SiC 15 μm is used. The A356 and A356 composite feedstock is shown in Figure 3.

- Stage 2: Thixoforming

Figure 4 shows the schematic of the experimental set-up. Slugs, 35 mm in diameter and 35 mm in height, were machined from the feedstock and used for thixoforming. The slugs were reheated by induction to 583 °C (approximately 0.4 liquid fraction by DSC), and then pressed in a heated SKD61 steel mold, using a ram speed of 0.016 m/s.

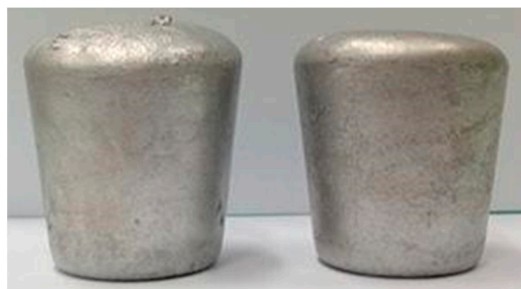

**Figure 3.** A356/SiC feedstock.

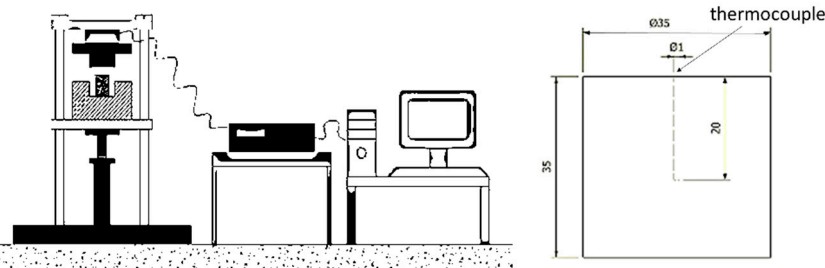

**Figure 4.** Schematic of the experimental arrangement for thixoforming a machined slug.

The Newtonian viscosity of a slurry (η, Pa·s), deformed between two parallel plates according to Laxmanan and Flemings [14], can be expressed by

$$\eta = \frac{2\pi h^5 P}{3v^2 (dh/dt)},$$ (1)

where *P* is the Load (N), *dh/dt* is the ram velocity (m/s), h is instantaneous height of the slug (m) at time *t*, and *v* is the slug volume ($m^3$). The slug under the test has an initial height of $h_o$ at *t* = 0.

The average shear rate, throughout the volume, is

$$\dot{\gamma} = \left| \frac{r}{2h^2} \left( \frac{dh}{dt} \right) \right|,$$ (2)

where *r* is the instantaneous radius of the deforming slug.

- Stage 3: Rheo-die-casting process

The A356/SiC-15% SiC of 15μm particle size feedstock was re-melted by induction. The A356/SiC slurry, was then transferred to the horizontal short sleeve of a die-casting machine (BD-125V4-T, Toyo machinery & metal Co., Ltd, Tokyo, Japan) for injection into the die at a temperature between 610 to 615 °C (corresponding approximately 0.90 to 0.95 liquid fraction). The ram speeds and injection pressures were higher than conventional aluminum alloy casting process; injection ram speeds of 3–4 m/s and pressures of 11–12 MPa. The mold was preheated to a temperature of 200 °C. A single injection produces 2-bar samples of $16 \times 15.6 \times 205$ $mm^3$, as shown in Figure 5.

The effect of T6 treatment on hardness of A356/SiC was examined. Specimens for T6 treatment are machined from the Rheo-die-cast rods at position no.1 and no.6, as shown in Figure 5, and heat-treated

to the T6 condition (solution treated at 540 °C for 1 h, followed by water quenching and ageing at 135 °C for 12 h).

Microstructural investigations of all samples were carried out using optical microscopy (Olympus BX60, Olympus optical Co., ltd, Tokyo, Japan) and Scanning Electron Microscopy (LEO model 1430, Carl Zeiss Microscopy GmbH, Jena, Germany). Brinell hardness testing is performed in accordance to ASTM E10-15a standard and the values presented are the average of three measurements.

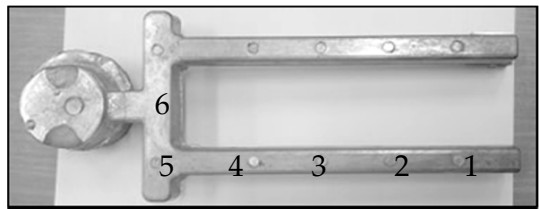

**Figure 5.** Example of rheo-diecast bars, including biscuit and runner and showing the 6 positions for samples used for metallography and hardness measurements.

## 3. Results and Discussion

### 3.1. Semi-Solid Stir-Casting

The microstructures of the stir-cast A356 alloy at various stirring speeds and times are shown in Figure 6. The microstructure consists of solid grains suspended in a liquid phase. An increase in stirring time resulted in increased grain size and more rosette-like grain morphology [15]. Increasing the stirring speed from 300 to 500 rpm resulted in increased grain size and reduction in globular features. It is clear that the average grain size of the 300 rpm-stirred sample is smaller than that of 500 rpm-stirred sample. Rapid stirring speed creates high shear rates in the slurry only near the blade, but lower shear rates near the crucible wall, due to centrifugal effects. These low shear rates may not be sufficient to shear and break down dendritic arms [16]. The hardness of stir-cast A356 at 300 rpm and 10 min stirring time was 72 HB. Therefore, the stirring condition of 300 rpm stirring speed and 10 min stirring time is an appropriate condition for fabricating A356SiC composite.

The microstructures of the A356/SiC (15%SiC-37μm) feedstock shown in Figure 7 reveal a uniform distribution of reinforcing particles in the A356 matrix. By using a rotor with 8 blades tilted 25 degrees to the vertical and a stirring speed of 300 rpm at continuous stir, from 680 °C (liquid phase) to approximately 615 °C corresponding to 0.9 liquid fraction for 10 min, SiC particles were successfully distributed in the matrix. Most of the reinforcing particles are found in the eutectic phase, and a small amount of particles were embedded in the solid phase; however, clusters of the reinforcing particles are also found in the eutectic phase. The hardness of the 15%SiC-37μm and 15%SiC-15μm are 74 HB and 52 HB, respectively. The increase in hardness of AMMCs appears to be caused by dislocations inhibiting the reinforcement phase. Low hardness of A356/15%SiC-15μm was due to the non-uniform distribution of carbide particles and gas porosity.

The microstructure at the center of thixoformed A356 is shown in Figure 8a. The microstructure consists of spheroidal grains without liquid entrapped in solid phase. This thixoformed structure appears to be more globular compared to the more rosette-like grains in the initial feedstock (Figure 6). The globulization of solid grains was due to increased shear rate at the beginning of the process. This non-dendritic structure with sufficient amount of liquid allows laminar flow of the slurry. However, although higher reheating temperatures, i.e., greater fraction liquid, and longer isothermal holding time improve globalization, the microstructures are susceptible to grain coarsening and increase in entrapped liquid in the solid grains which are prone to shrinkage porosity [17,18]. It is reported that hardness can be improved by decreasing the re-heating temperature of semi-solid A356 [19]. In addition, increase in cooling rate by cooling slope of a rheo-cast A356 alloy also produced near spherical structures, which enhance tensile strength compared to dendritic structures produced by

gravity casting [20]. Figure 8b,c shows thixoformed samples and microstructures at the center of a thixoformed slug of A356/15%SiC-37µm and A356/15%SiC-15µm, respectively, indicating increased particle size and decreased segregation of the solid phase. In addition, larger particles tend to inhibit the slurry flow. Some of the reinforcing phase did not flow with the liquid phase. Both composites appear to have more reinforcing phase at the center of the slug, while the periphery of the slug is mainly composed of the matrix phase. The remaining particles are mainly found at the eutectic phase between solid grains. Porosity is also found where 15µm-particles are clustered. This porosity appears to be caused by liquid segregation and localized cooling effects during solidification.

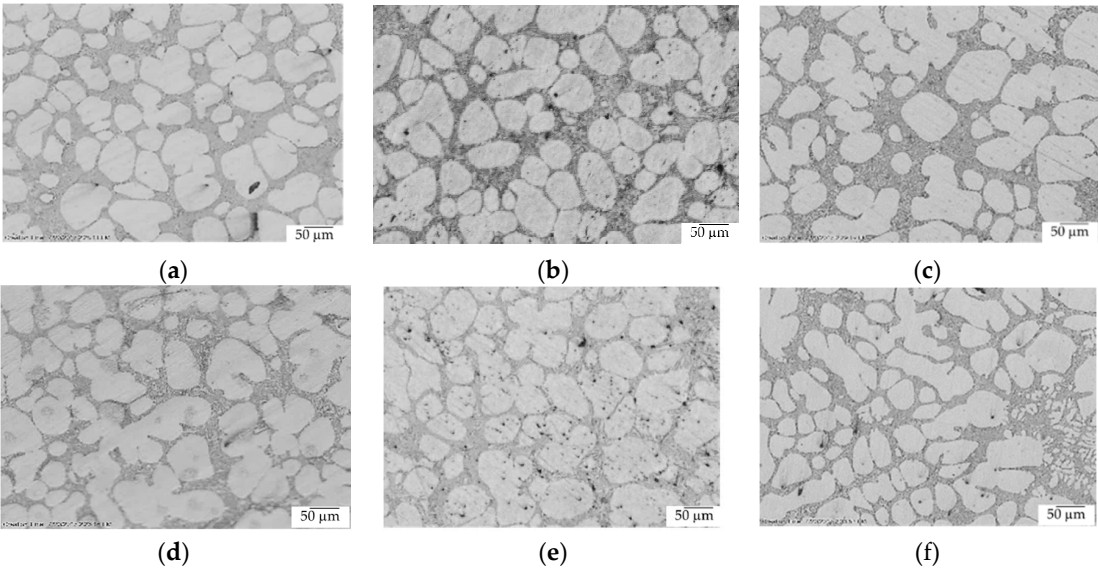

**Figure 6.** Optical micrographs of stir-cast A356 alloy. (**a**) 300 rpm, 5 min; (**b**) 300 rpm, 10 min; (**c**) 300 rpm, 15 min; (**d**) 500 rpm, 5 min; (**e**) 500 rpm, 10 min; and (**f**) 500 rpm, 15 min.

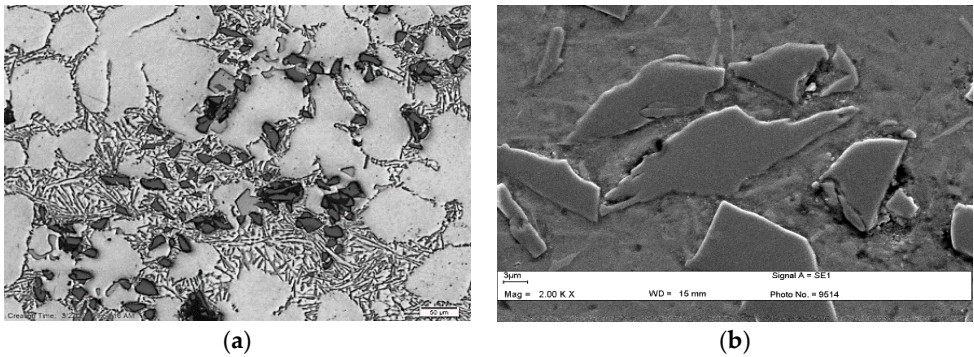

**Figure 7.** (**a**) Micrograph of mid-radius position of A356/SiC feedstock (15%SiC-37µm), and (**b**) SEM image showing SiC particles embedded in the eutectic phase.

Figure 9 shows a comparison of average viscosity against shear rate of the semi-solid A356 alloy and the A356/SiC composite. It can be clearly seen that the semi-solid A356 alloy and A356/15%SiC-15µm composite have quite similar values of maximum viscosity, but the A356/15%SiC-37µm composite exhibits much greater viscosity compared to the A356. An increasing viscosity results in lower flow-ability and increases the liquid-solid segregation. This is in agreement with the reduction in segregation of the reinforcement phase found in the thixoformed A356/15%SiC-15µm. However, viscosity reduces with shear rate, implying that A356/SiC exhibits shear thinning or non-Newtonian behavior. This is caused by the characteristic feedstock structure obtained, a relatively globular grain structure that accommodates the flow of the semi-solid composite. At the maximum stress, all phases

were compressed and particles and solid grains flow together with the liquid phase with increasing shear rate [21].

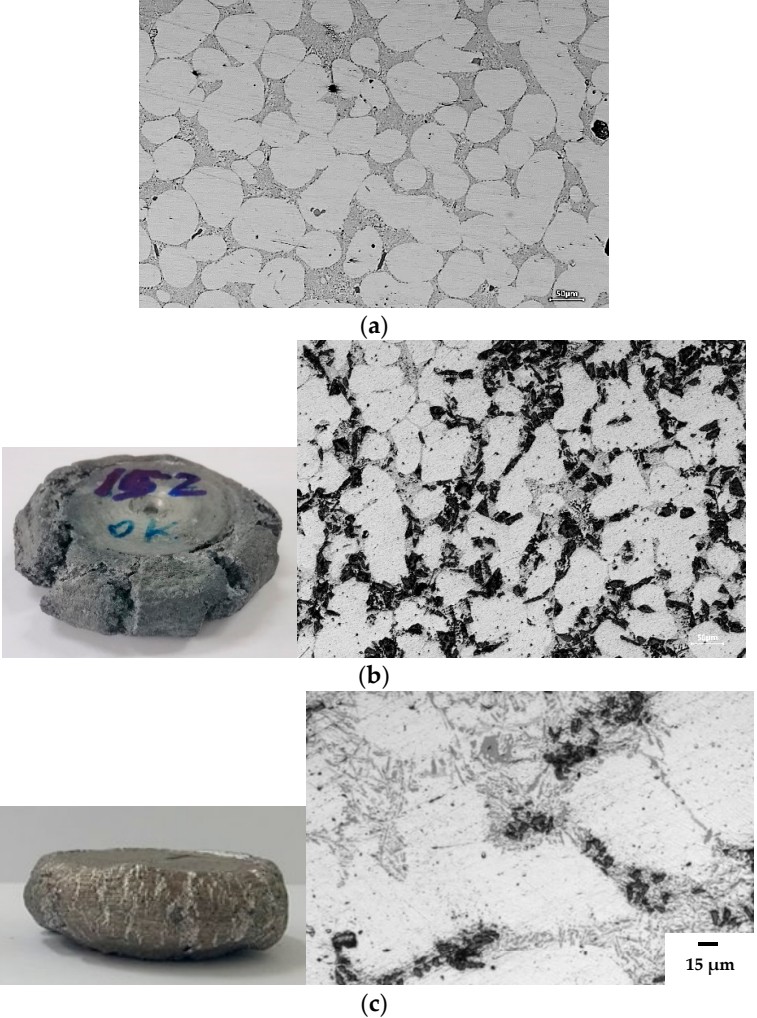

(a)

(b)

(c)

**Figure 8.** Microstructures after thixoforming of (**a**) A356; (**b**) A356/SiC feedstock (15%SiC-37μm); and (**c**) A356/SiC feedstock (15%SiC-15μm).

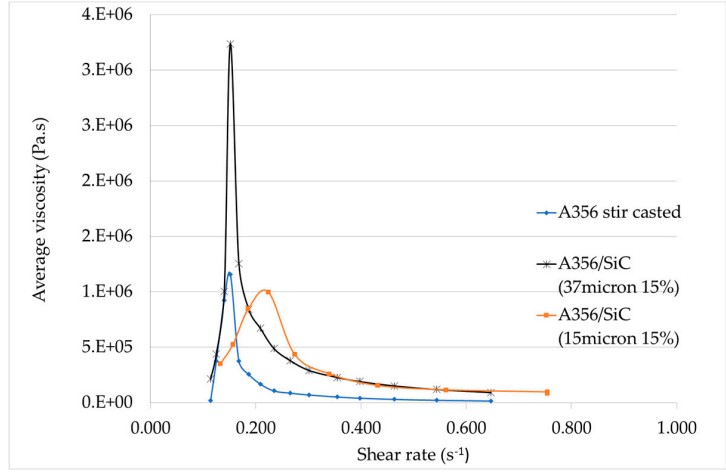

**Figure 9.** Average viscosities against shear rate of A356, A356/SiC feedstock (15%SiC-37μm), and A356/SiC feedstock (15%SiC-15μm).

### 3.2. The Rheo-Die-Casting Process

Examples of rheo-die-cast samples of A356/SiC(15%SiC-15μm) are shown in Figure 10. Due to the rheology change and longer flow cavity, the speed and injection pressure for the 0.9 fraction liquid A356/SiC slurry are set to be higher than that of conventional aluminum alloy. It was found that the rheo-die-cast samples exhibited uneven fill at the end of the part with injection conditions of 3 m/s speed and 11 MPa die pressure. When the speed and/or injection pressures were increased, the samples successfully and evenly filled the die cavities.

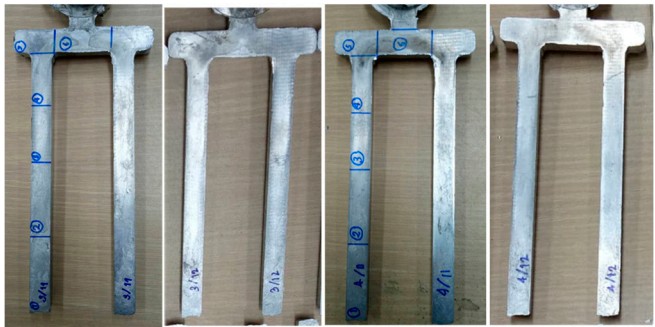

**Figure 10.** Rheo-die-cast samples at different speeds and pressures. From left to right: 3 m/s-11 MPa, 3 m/s-12 MPa, 4 m/s-11 MPa, and 4 m/s-12 MPa.

Cross-sectional microstructures along the bar length in positions 1 to 4 (illustrated in Figure 5) at various conditions are shown in Figures 11–14. The microstructures of the low speed, low pressure rheo-die-cast A356/SiC bar revealed an uneven dispersion of reinforcing SiC particles in the A356 matrix along the part, whilst increases in both speed and injection pressures improved a distribution of reinforcement phase. There was no evident shrinkage porosity or gas entrapment, despite some gas porosity found in the feedstock. This was because the viscosity of semi-solid composite slurries is higher than liquid metal, with the slurry flow being laminar, making air entrapment difficult. When the speed increases from 3 to 4 m/s, the slurry travels faster and fully fills the die cavity, silicon carbide particles are better dispersed at the end of the part (positions 1 and 2), as shown in Figure 11a,b, Figure 12a,b, Figure 13a,b, Figure 14a,b and Figure 15a,b. When the injection speed is constant, the pressure affects the distribution of silicon carbide particles. In Figure 15, microstructures of position 4 from all rheo-die-cast parts shows improved carbide distribution with increasing speed and injection pressure.

The hardness plots along the length of rheo-die-cast parts are shown in Figure 16. The low-speed and low-pressure condition produces non-uniform distribution of phases, as seen in Figures 11–15 resulting in non-uniform hardness. The most uniform hardness along the bar length is obtained from a sample fabricated at 4 m/s speed and 12 MPa pressure. This is due to the uniform distribution of the reinforcing SiC particles in the A356 matrix resulting in uniform hardness along the bar length.

Brinell hardness tests were performed after forming and T6 conditions, as shown in Figures 17 and 18. In the as-rheo-die-cast condition, the maximum hardness of a sample cast at 4 m/s speed and 12 MPa pressure was value of 82.16 HB. A sample cast with speed of 3 m/s, exhibited non-uniform hardness due to defects resulting from uneven slurry flow at the end of the part (position 1). It is clearly seen that injection speed affects the flow-ability of the slurry, while die pressure affects the density of the casting. The increasing hardness in the as-cast samples appears to be due to small, dispersed SiC particles interfering with the movement of high amounts of dislocations that are generated by thermally induced stresses arising from differences between the thermal expansion coefficients of matrix and reinforcement [22]. In thixoforming, the hardness of thixoformed slugs is lower than that of rheo-die-cast bars. This appears to be because the rheo-die-cast bars have lower amount of porosity and more uniform distribution of the reinforcing phase in the matrix.

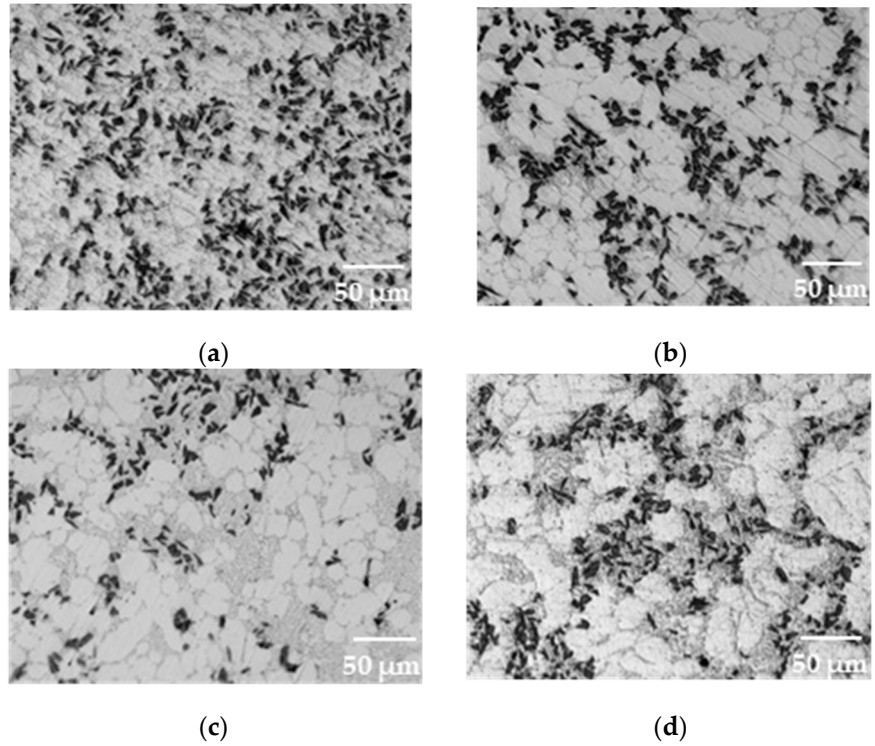

**Figure 11.** Cross-sectional microstructures of rheo-die-cast A356/SiC(15%SiC-15μm) bar at Speed = 3 m/s and Pressure = 11 MPa in (**a**) position no.1, (**b**) position no.2, (**c**) position no.3, and (**d**) position no.4.

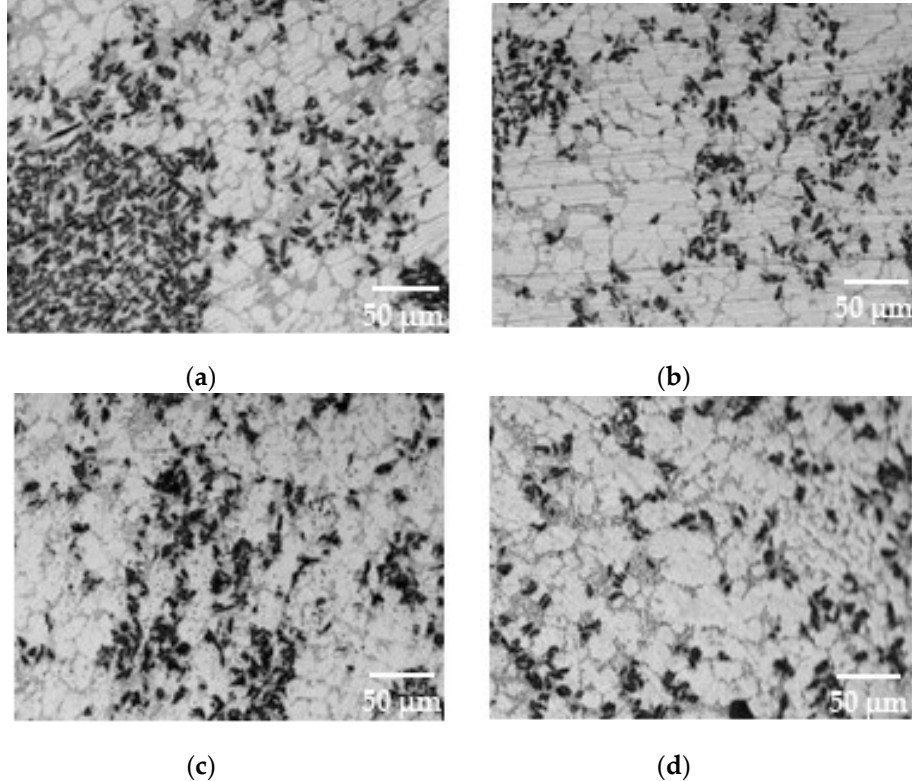

**Figure 12.** Cross-sectional microstructures of rheo-die-cast A356/SiC(15%SiC-15μm) at Speed = 3 m/s and Pressure = 12 MPa in (**a**) position no.1; (**b**) position no.2; (**c**) position no.3; and (**d**) position no.4.

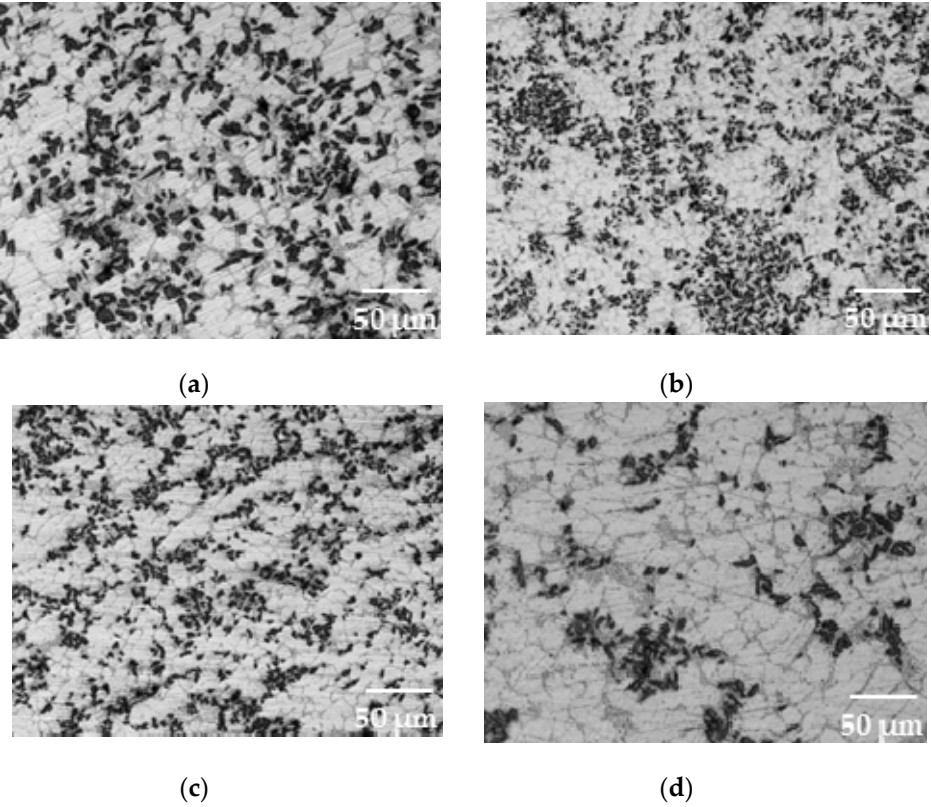

**Figure 13.** Cross-sectional microstructures of rheo-die-cast A356/SiC(15%SiC-15μm) at Speed = 4 m/s and Pressure = 11 MPa in (**a**) position no.1; (**b**) position no.2; (**c**) position no.3; and (**d**) position no.4.

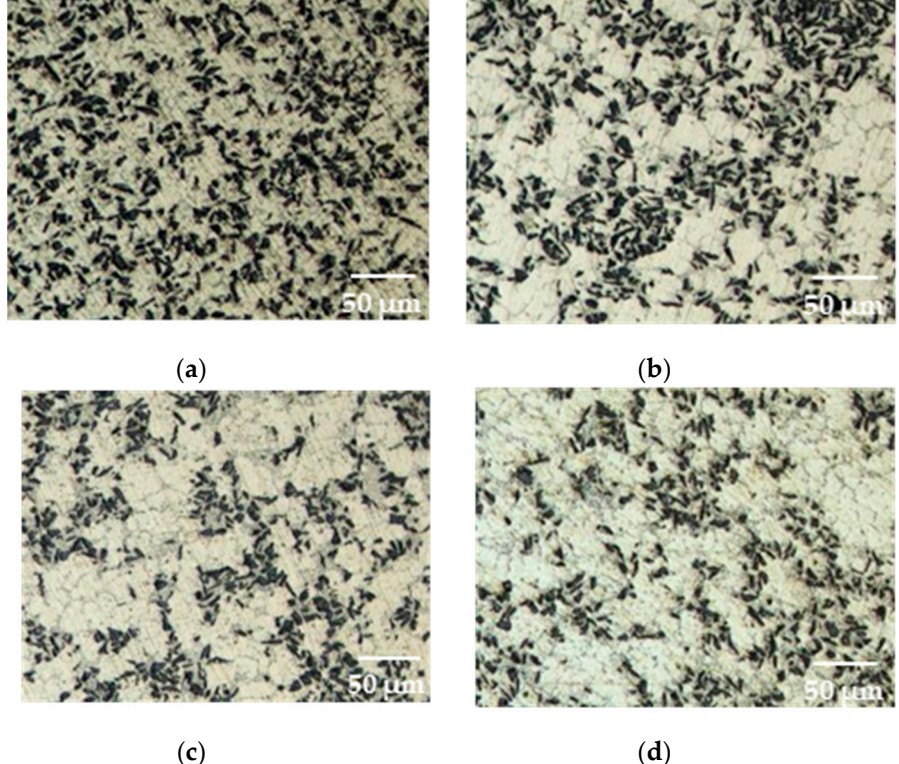

**Figure 14.** Cross-sectional microstructures of rheo-die-cast A356/SiC(15%SiC-15μm) at Speed = 4 m/s and Pressure = 12 MPa in (**a**) position no.1; (**b**) position no.2; (**c**) position no.3; and (**d**) position no.4.

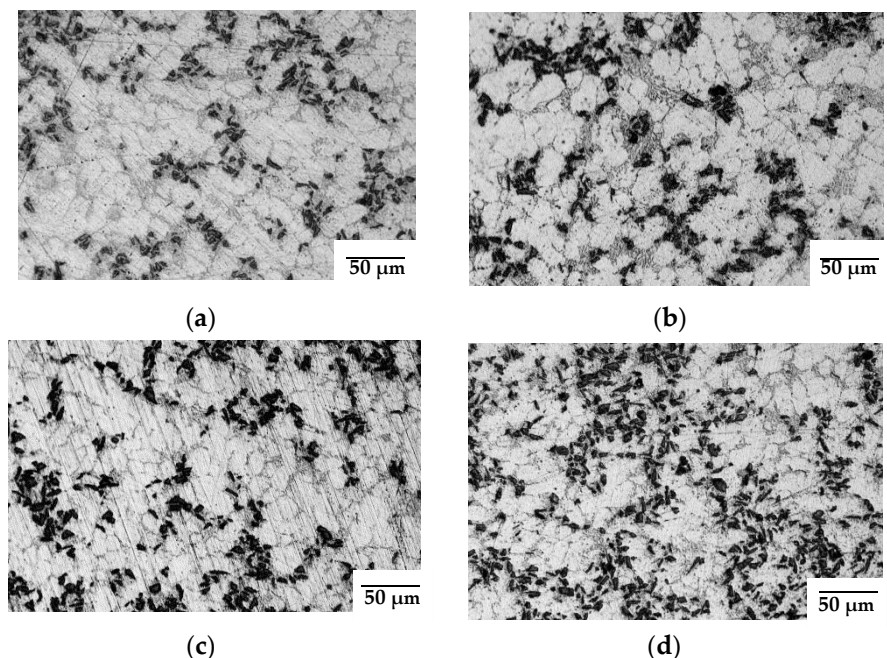

**Figure 15.** Cross-sectional microstructures in position no.4 at various injection conditions of rheo-die-cast A356/SiC(15%SiC-15μm) of (**a**) Speed = 3 m/s, Pressure = 11 MPa; (**b**) Speed = 3 m/s, Pressure = 12 MPa; (**c**) Speed = 4 m/s, Pressure = 11 MPa; and (**d**) Speed = 4 m/s, Pressure = 12 MPa.

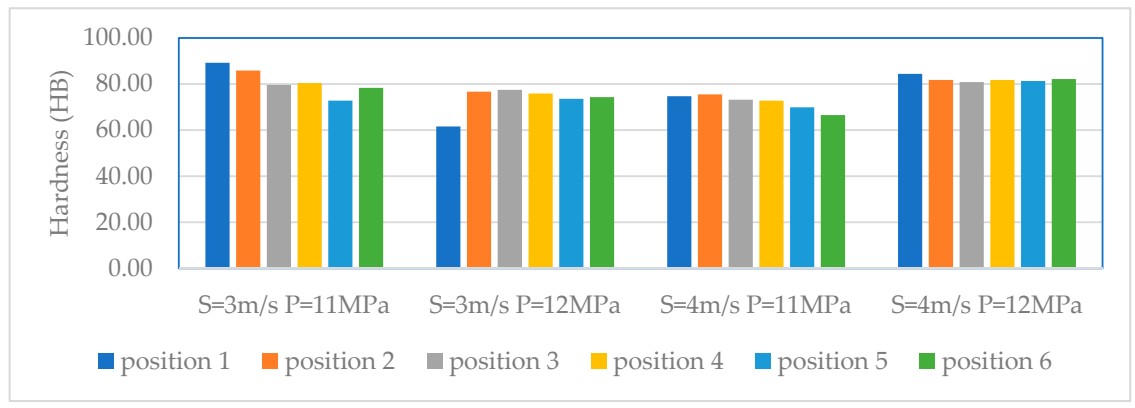

**Figure 16.** Hardness of rheo-die-cast bars.

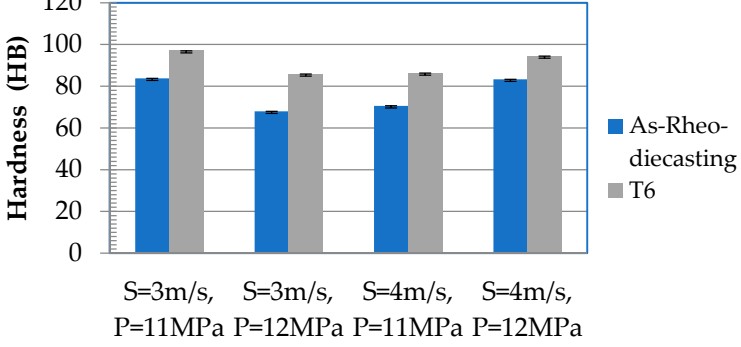

**Figure 17.** Average hardness of rheo-die-cast A356/SiC(15%SiC-15μm) of position 1 and 6.

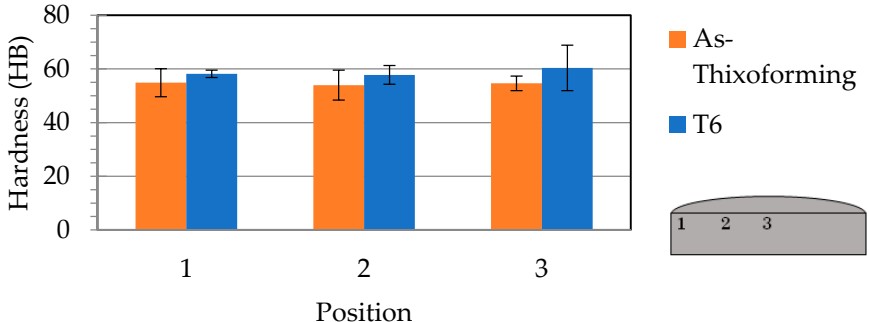

**Figure 18.** Average hardness of thixoformed A356/SiC(15%SiC-15μm).

After T6, hardness values increased by 15.6% and 7.3% in the rheo-die-cast and thixoformed samples, respectively. This increase in hardness is due to the formation of $Mg_2Si$ precipitates during ageing inhibiting dislocation movement. The presence of particulates may modify vacancy contents, facilitate diffusion of solutes and provide additional sites for the heterogeneous nucleation of precipitating phase [22]. In addition, the SiC can modify the morphology of eutectic phase. The morphology of Al-Si eutectic formed by rheo-die-casting and thixoforming also are refined and modified from needle-like to more globular, as shown in Figures 19 and 20.

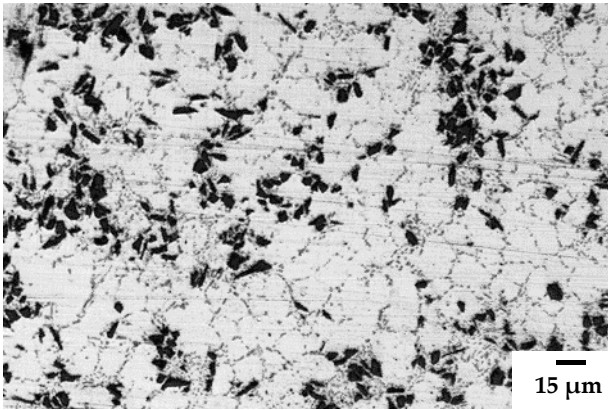

**Figure 19.** The microstructure of rheo-die-cast A356/SiC(15%SiC-15μm) in T6 condition.

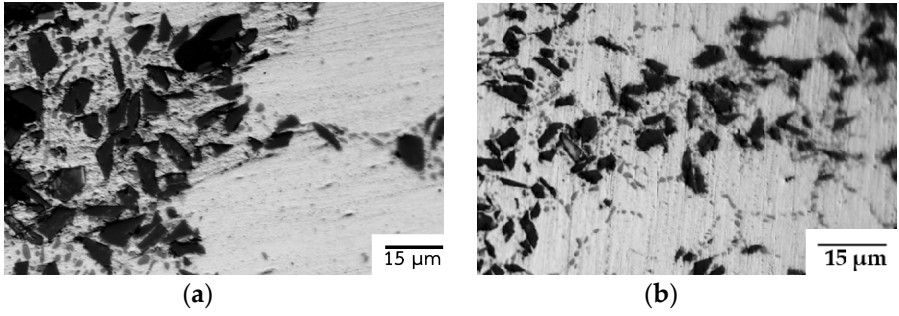

**Figure 20.** Micrographs showing modification of the Al-Si eutectic phase of (**a**) a thixoformed sample and (**b**) a rheo-die-casted sample.

A comparison between thixoforming and rheo-die-casting of A356/SiC(15%SiC-15μm) is shown in Table 2. The results clearly indicate that greater speeds, pressures and cooling rates in the rheo-die-casting process improve the uniformity of microstructure and hardness of the composites and

that a reasonable process window for thixoforming is possible by increasing the speed and/or pressure to increase shear rate and flow-ability of the more viscous composite.

**Table 2.** Comparison of thixoforming and rheo-die casting of A356/SiC (15%SiC-15μm).

| Criteria | Semisolid Process | |
| --- | --- | --- |
| | **Rheo-Die-Casting** | **Thixoforming** |
| Fraction liquid<br>Pressure<br>Speed | Approximately 0.9<br>11 and 12 MPa<br>3 and 4 m/s | Approximately 0.4<br>6 MPa<br>0.016 m/s |
| Microstructure | - Uniform distribution of SiC in liquid phase<br>- Uniformity increases with increasing speed and pressure | - Some clusters of SiC in eutectic phase<br>- Porosity caused by liquid segregation |
| Hardness | - As rheo-cast = 83.31 HB<br>- 15.6% increasing after T6 | - As thixoformed approximately 60.11 HB<br>- 7.3% increasing after T6 |

## 4. Conclusions

(1) A stir-casting process in the semi-solid state, with a stirring speed of 300 rpm and stirring time of 10 min, was successful in producing non-dendritic A356 alloy and A356/SiC composite feedstock with good distribution of particulates in the matrix. Shearing promoted near globular and rosette-like solid grains in the liquid phase, which is suitable for subsequent semisolid forming. After thixoforming at 0.4 fraction liquid, the microstructure of A356 alloy appeared to have more spheroidal grains without liquid entrapped in a solid phase. The slurry at 0.4 liquid fraction exhibited low viscosity and non-Newtonian behavior.

(2) At the same amount of particle loading, an increase in reinforcement particle size increases the viscosity of the composite. An increasing viscosity results in lower flow-ability and increases the liquid-solid segregation.

(3) The hardness of the matrix phase in the T6 condition increased with the amount of reinforcement phase. Composites with small-reinforcing particles exhibited greater hardness at higher amounts of particles, while composites with large reinforcing particles exhibited high hardness.

(4) In thixoforming, the feedstock thixoformed at 583 °C, equivalent to 0.4 fraction liquid, and viscosity is reduced with shear rate, implying that the A356/SiC exhibits shear thinning or non-Newtonian behavior. This is caused by the characteristic relatively globular grain structure of the billet that accommodates the flow of the semi-solid composite slurry.

(5) Injection speed clearly affects the flow-ability of slurries, while die pressure affects the density of the castings. In the rheo-die-casting process, the A356/SiC slurry is die-cast at 610–615 °C (approximately 0.8–0.9 fraction liquid). The most suitable die casting condition for semi-solid A356/SiC composites appears to be that of 4 m/s speed and 12 MPa die pressure, respectively. This condition allows an even flow and uniform distribution of reinforcing particles in the A356 matrix. There was no evident shrinkage porosity or gas entrapment in the rheo-die-cast parts. This condition also exhibited uniform hardness, resulting from the uniform microstructure of the sample, as compared to other conditions.

**Author Contributions:** Conceptualization, S.T.; methodology, S.T.; formal analysis, S.T.; investigation, C.P. and P.S.; writing—original draft preparation, S.T.; writing—review and editing, S.T.; All authors have read and agreed to the published version of the manuscript.

**Funding:** This research was funded by Supply Chain and Logistics System Research Unit, Department of Industrial Engineering, Faculty of Engineering, Khon Kean University.

**Acknowledgments:** The authors would like to express their gratitude to the Faculty of Engineering, Khon Kaen University and U.M.C Die Casting Co., Ltd. for the technical support.

**Conflicts of Interest:** The authors declare no conflict of interest. The funders had no role in the design of the study; in the collection, analyses, or interpretation of data; in the writing of the manuscript, or in the decision to publish the results.

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
