# Peer review of "Thixoforming and Rheo-Die-Casting of A356/SiC Composite"

_metals, doi:10.3390/met10020251_

Round 1

Reviewer 1 Report

Why do not you show mechanical properties of similar parts obtained by rheocasting and thixoforming from A356 alloy without SiC additions? It should be done to prove purpusefulness of this research. The control parameters of rheocasting and thixoforming processes are unclear. Could you write how did you control these experiments? By piston velocity or applied piston pressure? How did you detect SiC phase on Figure 5b?

Author Response

Reviewer 1

Why do not you show mechanical properties of similar parts obtained by rheocasting and thixoforming from A356 alloy without SiC additions? It should be done to prove purpusefulness of this research.

The aims of this work is focus on the rheocastability and thixiformability of composite materials. Since the thixoformabiity of semisolidA356 has been  intensively done by other researchers.

The control parameters of rheocasting and thixoforming processes are unclear. Could you write how did you control these experiments? By piston velocity or applied piston pressure?

The speed in thixoform operation is measured by the ram velocity during forming and in rheo-die casting, the speed is an injection speed controlled by the piston speed of a HPDC machine.

How did you detect SiC phase on Figure 5b? 

Author added an indication of SiC in the Figure.

Reviewer 2 Report

The work reports an investigation on the distribution of SiC particles in an Aluminum alloy (A356) produced by rheo-casting process. However, the manuscript does not meet the necessary requirements to be published; I suggest: 1) to improve the english level; 2) the better explain the experimental procedure; 3) to add some extra analyses in addition to hardness test results and optical microstructure investigations; 4) to extend the discussion of the results.

Author Response

Reviewer 2

The work reports an investigation on the distribution of SiC particles in an Aluminum alloy (A356) produced by rheo-casting process. However, the manuscript does not meet the necessary requirements to be published; I suggest:

1) to improve the english level;

The English level has been read and improved by an english native speaker.

2) the better explain the experimental procedure;

Author add figures to schematic of experiments and explains stages of experiments.

3) to add some extra analyses in addition to hardness test results and optical microstructure investigations;

Author added more explanation on DSC curve, and microstructures.

4) to extend the discussion of the results.

Author added and extened the discussion of added microstructures and hardness results.

Reviewer 3 Report

What are the advances of your research beyond the state of the art? in other words what is originality of your work?

As an example have read these papers?

https://www.sciencedirect.com/science/article/pii/0921509394096805

Author Response

What are the advances of your research beyond the state of the art? in other words what is originality of your work?

The aims of this work is focus on the rheocastability and thixiformability of composite materials. Therefore, first we develop the feedstock production using stir casting then the flow and viscosity of composites were measured to evaluate the rheocastability and thixiformability of composite materials. Also, the effect of speed and pressure on carbide distribution of MMCs are analyszed in order to evaluate whether the rheo-die casting of MMCs can perform using a conventional high pressure die casting without a machine modification.  

As an example have read these papers? https://www.sciencedirect.com/science/article/pii/0921509394096805

Authors have read and cited this article.